# The Nutraceutical Properties of *Rhus coriaria Linn*: Potential Application on Human Health and Aging Biomedicine

**DOI:** 10.3390/ijms24076206

**Published:** 2023-03-25

**Authors:** Anna Calabrò, Mattia Emanuela Ligotti, Giulia Accardi, Danila Di Majo, Calogero Caruso, Giuseppina Candore, Anna Aiello

**Affiliations:** 1Laboratory of Immunopathology and Immunosenescence, Department of Biomedicine, Neurosciences and Advanced Diagnostics, University of Palermo, 90134 Palermo, Italy; 2Department of Biomedicine, Neurosciences and Advanced Diagnostics (BIND), University of Palermo, 90127 Palermo, Italy; 3Postgraduate School of Nutrition and Food Science, University of Palermo, 90100 Palermo, Italy

**Keywords:** aging, health, immunoceuticals, nutraceuticals, phytochemicals, Sicilian Sumac

## Abstract

*Rhus coriaria Linn* is a little plant growing in the Mediterranean basin, including Sicily, where it is known as Sicilian Sumac. Since antiquity, it has been used as a medicinal herb, considering its pharmacological properties and its recognized anti-inflammatory, antioxidant, and antimicrobial effects. Multiple studies have highlighted that the beneficial properties of Sumac extracts depend on the abundance of phytochemicals such as polyphenols, fatty acids, minerals, and fibers. Despite its wide use as a spice, the literature on Sumac effects on humans’ health and aging is still scarce. Considering its great nutraceutical potential, Sumac could be used to treat age-related diseases such as those in which the inflammatory process plays a crucial role in manifestation and progression. Thus, Sumac could be an interesting new insight in the biomedical field, especially in aging biomedicine.

## 1. Introduction

*Rhus coriaria Linn* (*R. coriaria* L.) is a little plant commonly known as Sumac, originating from the subtropical and temperate region of the world, particularly diffused in the Middle East and the Mediterranean basin. The name “Sumac” derives from the Arabic and Syriac word “Sumaga” or “Summaqua”, which precisely means “dark red”, evoking the color of the mature fruits [1]. Indeed, its peculiar characteristics are the presence of pinnate leaves and greenish-white flowers harvested in cobs, while the fruits are reddish-brown drupes [2]. The plant reaches maturity at three years of age, giving rise to seeds that follow the formation of panicle inflorescences. When the flowering is complete, the small fruits assume a spherical shape; their color is initially green, then bright red until full maturity is reached, and then they become darker before withering. The harvesting time is August–September, which is when the fruits reach complete ripeness and appear dark red [3,4]. The composition of its extracts and essential oils determines its organoleptic properties and its employment, i.e., the lemonade taste is given by the content in limonene, the sour taste of fruits is due to malic acid, and the spiciness is given by caryophyllene [2,5,6].

Its use and trade have taken place since ancient times and in different fields of application. In the Middle East regions, Sumac is usually harvested to obtain powder from different parts of the plant for the food and cosmetics industry, and as a dressing for seasonal foods such as a folk salad of onions, kebabs, yogurt, rice, and hummus [6]. In the Mediterranean basin, it is used as a spice to flavor meat, fish, and salad, together with olive oil and vinegar. Syriac Sumac and Chinese Sumac were employed in food manufacturing because of their antimicrobial and anti-inflammatory properties, which are due to the composition of essential oils rich in citric acid [1,6,7].

In Italy, Sumac is widespread in Western and South-Western Sicily, and is termed “Sicilian Sumac” to distinguish it from the plants of other regions, which differ in the extract’s composition and organoleptic properties [8]. Sicilian Sumac has been utilized as a colorant due to its tannins content, which explains the old use of Sumac to tan leather. Sumac is also widely used as a medicinal herb [8]. Ancient populations in the Middle East applied the powder obtained from Sumac leaf or fruits to cure illnesses such as fever and gut and liver diseases, as well as alter lipid and glucose homeostasis, anticipating studies about the composition of its extracts, and showing a possible application of Sumac in clinical practice.

Sumac extracts are suggested to explicate a large spectrum of actions, with anti-bacterial, anti-viral, anti-inflammatory, and antioxidant effects, thanks to the properties of its phytochemicals. Their use could have a low economic impact, in line with the easy availability of the material, inexpensive cultivation techniques, and short- and low-impact processing.

The discovery of the therapeutic properties of Sumac aroused the interest of researchers across the last 10 years. Sumac was hypothesized as a “nutraceutical” and its application in functional food production was encouraged [9]. Moreover, in the light of the properties of Sumac regarding the immune field, it is possible to suggest that Sumac compounds also have immunoceutical characteristics, acting on the immune system and favoring the activation or inhibition of immune system cells to respond to pathogenic insults [9].

The research of new phyto-therapeutic molecules and their possible applications in different biomedicine fields is growing to resolve or treat pathological conditions that could overcome drug resistance or sensitivity issues, improve the environmental impact, and reduce the economic drift versus more expensive pharmaceutical compounds [10]. For these reasons, the object of this review will be describing the nutraceutical and immunoceutical properties of Sumac—in particular the Sicilian one—in light of two main changes of our times: the increasing mean age of the world population according to the rising incidence of age-related diseases, and the development of new infective agents impacting the enhanced spectrum of infectious diseases, all in order to suggest new promising applications for this plant.

## 2. Sumac and Sumac Extracts Composition

The analysis of Sicilian Sumac revealed a prevalent composition of phytochemical compounds in the Sumac fruits, such as organic and phenolic acids, flavonoids, hydrolysable tannins, and anthocyanins. The principally identified flavonoids of methanol/aqueous extracts in Sicilian Sumac are quercetin 3-O-galactoside, kaempferol 3-O-glucoside, quercetin, and myricetin 3-O-hexoside. Among the organic acids, the main representatives are gallic acids and pentagalloyl-hexoside [11]. The volatile compounds of Sicilian Sumac determine the composition of essential oils and are prevalently derived from monoterpenoids and sesquiterpenes such as β-caryophyllene and α-pyrene, respectively, and from non-terpenes compounds such as aldehydes [7,12]. The aroma profile of methanol/aqueous Sicilian Sumac extracts is described by aromatic and volatile compounds, including non-terpenes. Furthermore, fibers and unsaturated fatty acids, such as palmitic acid, linolenic and α-linoleic acids (omega-3 and -6), can be found. Potassium, calcium, magnesium, and phosphorus are the major mineral components in Sicilian Sumac [11]. Exhaustive information about the composition in vitamins is lacking, although some scientific works analyzing Syrian Sumac water extracts described a main content of pyridoxine, ascorbic acid, thiamine, and riboflavin [5,13].

Information on the composition of Sumac extracts is obtained by carrying out extractions using different types of solvents, including methanol, ethanol, acetone (alcoholic extracts), and water (aqueous extracts). It is noticed that extracts with different compositions could be obtained based on the chemical characteristics of the extraction solvents and on the different parts of the plant used. Alcoholic extracts (methanol and ethanol extracts) of Sumac leaves are rich in quercetin, myricetin, kaempferol, and gallic acid [8]. Ethanol/water extracts show a greater content of polyphenols in leaves than in fruits. Moreover, carbohydrates, essential oils, flavonoids, and tannins, which are the most interesting nutraceutical components of the plants, are found in leaf extracts. Methanol extracts from Sumac fruits contain 78 different hydrolyzed tannins, 59 flavonoids, 9 anthocyanins, and 40 other molecules, while aqueous extracts are rich in organic acids such as malic, citric, fumaric, and tartaric [8,14]. The seed composition is like that of leaves and fruits regarding hydrolysable tannins, flavonoids, anthocyanins, organic acids, and steroids [8,14]. In addition to the chosen solvents, it is necessary to consider the methods applied to obtain the extracts, such as the concentration of the solvent, the temperature used during the extraction, the ratio between solvent and Sumac plant powder, and the size of the particles obtained.

See Figure 1 for a summary of the main phytochemicals present in the different parts of the plant, and Table 1 and Table 2 for a detailed analysis of the extract composition based on the solvent, methods used for the extraction, and analysis of Sumac extracts composition.

The climatic conditions of the place where the Sumac grows, and the harvesting period are other factors that could influence the composition of the extracts [2]. Generally, Sumac grows in regions rich in calcareous soil, with chalky-sulfurous formations and a temperate climate. The study of volatile compounds in Sicilian Sumac from different Sicilian areas, such as Trapani, Ragusa, and Palermo, shows a different amount of monoterpene hydrocarbons and oxygenated monoterpenes due to different soil compositions and climatic conditions [2,3,4]. The harvesting time is important to obtain mature fruits rich in bioactive compounds, so the cut height of the plant and the drying temperature play an important role in avoiding pests and quality alteration. Thus, to ensure the production of extracts that both reflect the properties of Sumac and are rich in nutraceuticals, it is essential to establish the correct procedures, from cultivation to the setting up of the extraction protocol. Several studies showed that the properties of Sumac extracts depend both on the polyphenols’ content, which has a role in the reduction of lipidic absorption due to the high resin-binding capacities and greater anti-inflammatory properties, and on the tannins, which contribute to the antioxidant properties due to their action against the xanthine oxidase, which is involved in cholesterol metabolism [14,19,20].

## 3. Antioxidant, Anti-Inflammatory and Immunomodulatory Properties of Sumac

Oxidative stress and inflammation are correlated with the deterioration of health conditions and the exacerbation of diseases. Cell levels of reactive oxygen species (ROS) are regulated by the balance between their generation and detoxification. Scavenger systems fail to neutralize ROS and this leads to accumulation of them in cells. This condition is influenced by the disruption of cell homeostasis following environmental stressors, such as drug consumption and impaired clearance. Natural compounds such as polyphenols have a central role in the contrast of oxidative stressors and favor the scavenger activity against ROS [21]. The reduction of oxidative stress slows the overproduction of ROS and other radical elements, typical of different age-related diseases such as diabetes, skin injuries, obesity, and cardiovascular diseases [22,23].

The ability of Sumac extracts to avoid food deterioration due to oxidative process is well known. To date, it has been learned that the ability to contrast the peroxides and ROS presence depends on the abundance of polyphenols and flavonoids and, in particular, tannins in its extracts. Therefore, water Sumac extracts, at the concentration of 30 μg/mL, showed higher anti-oxidative effects than ethanol ones thanks to the greater percentage of phenols (63 μg GAE/mg) and flavonoids (0.6 μg Quercetin Equivalents/mg), determined by a colorimetric assay [24]. Tannins and their derivatives, predominant in water extracts, are rich in hydroxyls and carboxyl groups that permit them to interact with free radicals, donating electrons to oxidized molecules and interrupting radical chain reactions. Studies in vitro and in vivo have shown a higher antioxidant activity of alcoholic and water extracts at a range of concentrations from 75 μg/mL to 200 mg/mL, than that of vitamin C against nitroxide oxide, hydroxyl radical, and peroxidation [5,8,25]. This was demonstrated for skin microvascular endothelial cells, used as a model of dermal Ultraviolet ray type A (UVA)-induced damage, and irradiated with UVA. The use of 10 or 25 μg/mL of alcoholic extracts determined not only a reduction of ROS, maybe through the modulation of superoxide dismutase and catalase, but also the arrest of DNA damage eventually prompted by oxidative stress [26]. There was also evidence of the successful use of gallic acid, one of the principal components of Sumac extracts, in the reduction of H_2_O_2_-induced DNA damage in lymphocytes, after the administration of 3 g/day, for 3 days, of ethanolic Sumac extracts [27]. Flavonoids confer a strong anti-oxidative effect, acting on xanthine oxidase, cyclooxygenase (COX), lipoxygenase (LOX), and nitric oxide synthase (iNOS) [5]. Kaempferol, belonging to flavonoid compounds, acted on oxidative stress at the optimal concentration of 25µM, increasing Nuclear factor erythroid 2–related factor 2 gene expression [28]. The involvement of oxidative stress is also evident in metabolic diseases such as obesity and diabetes, in which the action of 3 g/day of ethanolic Sumac extracts consisted in the inhibition of pancreatic lipase, the modulation of glutathione S-transferase (GST) in the liver and of isozymes GST-α and GST-π in the plasma of human subjects [5,27]. Finally, the gallo-tannins of Sumac leaf extracts at the concentration of 150–500 µg/mL showed the capacity to gain an athero-protective role, reduce blood pressure, and reduce urea nitrogen level [29].

Inflammation also contributes to a wide range of diseases, including age-related ones. Inflammation is an organism’s normal response to an insult that may be caused by environmental chemicals, injuries, pathogens, and radiation. The clinical manifestations are redness, swelling, heat, and pain, mediated by cells which participate in the inflammatory process and by molecules released at the site of damage. Inflammation is associated with an increased level of pro-inflammatory cytokines, such as Interleukin (IL)-6, Interferon (IFN)-γ, Tumor Necrosis Factor (TNF)-α, acute phase proteins, as well as C reactive protein (CRP) [30]. Studies in vivo and in vitro revealed an important contribution of Sumac in reducing the inflammatory status and inducing the decrease of pro-inflammatory cytokines and CRP by administering 3 g/day of pure Sumac powder for 3 months [30]. Experiments investigating the anti-inflammatory action of Sumac put into evidence a possible modulation of the nuclear factor kappa-light-chain-enhancer of activated B cells (NF-κB) pathway, which guides the release of pro-inflammatory cytokines. NF-κB expression is linked to inflammatory status, and its inhibition could affect the onset of inflammatory diseases. In inflammation models of TNF-α-stimulated keratinocytes, *Helicobacter Pylori*-stimulated gastric epithelial cells, and Lipopolysaccharide (LPS)-stimulated calf synoviocytes, a modulation of the production of IL-8 at the half-maximal inhibitory concentration (IC50) of 3.15 ± 1.14 and 6.61 ± 0.55 μg/mL, for IL-18 and IL-1β, respectively, has been shown after treatment with the alcoholic extract of Sumac fruits at the half-maximal lethal concentration (LC50) of 0.42 μg/mL and 15–16% of yield of extracts, and also through the inhibition of the NF-κB signaling pathway (IC50 11.48 ± 0.21 μg/mL and 18.51 ± 0.08 μg/mL) [5,17,31,32]. This property seems to depend on the action of phenols and terpenoids found in Sumac extracts. Experiments on BV-2 microglia cells opened to the use of alcoholic and water extracts of Sumac at the concentration range of 25–50 μg/mL to reduce the oxidative stress and the inflammation also in neurodegenerative diseases through NF-κB modulation, suppressing TNF-α, iNOS and COX-2 expression, and increasing IL-10 expression [33]. Quercetin and kaempferol, important components of Sumac extracts, act on NF-κB, causing an increased production of IL-10 through the inhibition of TNF-α and IL-1β production and down-regulating the expression of IL-6 and IL-18. In addition, quercetin may regulate the expression of adhesion molecules and the production of metalloproteinases, blocking histamine production and reducing pro-inflammatory enzymes such as COX, LOX and iNOS [9].

The NF-κB pathway is also involved in the modulation of immune cell responses, which have a leading role in inflammatory response. In detail, the possibility to modulate macrophage activity is well known both in vitro and in vivo. In mice models, the extracts of *Rhus verniciflua Stokes*, a plant belonging to the Sumac family, at concentrations of 20 mg/mL or 100 mg/mL and at doses of 200 mg/Kg/day for 10 days, inhibited the expression of TNF-α and IL-6, and influenced the capacity of monocytes to differentiate in macrophages, increasing the expression of IL-12 and major histocompatibility complex (MHC) class II molecules, and also involving the extracellular signal-regulated kinase 2-mitogen-activated protein kinase signaling pathway [34]. Additionally, quercetin (25–50 µM) and kaempferol (20 µg/mL) modulate the activation and maturation of dendritic cells, regarding the antigen uptake capacity, MHC expression, and production of cytokines [9,35,36]. Furthermore, through the monocytes’ action, quercetin favors the polarization of the immune response toward T helper (Th) 1 response, inducing the Th1-producing IFN-γ at the concentrations of 25 μM and 50 μM, and inhibiting Th2-producing IL-4 at the concentrations of 5.25 and 50 μM [37]. Moreover, Sumac omega-3 fatty acids can upregulate the activation status of macrophages, neutrophils, T and B cells, dendritic cells, natural killer cells, mast cells, eosinophils, and basophils, as reported by the different studies. They regulate the activity of neutrophils and increase the activation of T cells by antigen presenting cells [9].

Thus, the immunoceutical properties of Sumac extracts and their role in the modulation of the immune response are important to investigate, suggesting potential applications in the pharmaceutical preparation of vaccines and immune-based drugs to enhance their response.

## 4. Antimicrobial and Antifungal Effects of Sumac

Recent studies have explored the antimicrobial effects of natural compounds as a novel approach to limiting the phenomenon of antibiotic resistance.

A study on the action of the antimicrobial effects of Iranian Sumac was developed using different bacteria strains. The study regarded both Gram-positive bacteria, i.e., *Bacillus cereus*, *Lysteria monocitogenes*, *Staphylococcus aureus,* and Gram-negative bacteria, i.e., *Escherichia coli* and *Salmonella enteritis.* It was shown that the action of water extracts at concentrations of 0.1%, 0.5%, 1.0%, 2.5%, and 5.0% (*w*/*v*) is independent from the stage of maturation of the plants but depends on the concentration of phytochemicals. Moreover, by increasing the concentration of the compounds (from 0.5% to 5.0%) in the extracts, a major antimicrobial activity was discovered. The enhanced susceptibility of Gram-positive bacteria to Sumac has been also demonstrated with the use of ethanol extracts, with a major contribution from malic and citric acids. Instead, Gram-negative resistance could depend on the polarity of the membrane, composed of LPS, which prevents the adsorption of Sumac extract [38].

Tests of different Sumac phytochemicals on *Enterobacteriaceae* multidrug resistance strains showed the best antimicrobial activity for the methanol and ethanol extracts with a Minimum Inhibitory Concentration of 9.37 μg/mL, with a major action of the first one because of its higher content in phenolics and flavonoids [11].

The rise in bacterial-type infections due to the increased resistance to antibiotics goes hand in hand with an augmented incidence of viral-type infections, which are often the cause of epidemics and pandemics, favored by environmental and anthropological factors. Viral infections are more difficult to eradicate due to the mutagenicity of some viruses and the shortage of antiviral drugs that can be readily used against different species. Several in vitro studies showed the ability of Iranian Sumac extracts to have antiviral activity. For example, bioflavonoids in the Sumac seem to have effects against influenza viruses (half maximal effective concentration (EC50) values of 2.0 µg/mL and 0.2 µg /mL), Human Immunodeficiency Virus (HIV)-1 reverse transcriptase (aqueous extracts of a plant of Sumac family at IC50 of 15 μg/mL and ethanolic extracts at IC50 of 26 μg/mL), and Herpes viruses (EC50 of about 8.6 µg /mL; IC50 of 0.0005% and 0.0043% for Sumac extracts) [39]. The inhibition of the Hepatitis B Virus (HBV) in hepatoma cells by Sumac flavonoid compounds at the concentration of 30µM has been also demonstrated [40]. The mechanism of the inhibition of these viruses is still dubious. Sumac may act on the envelope of the viruses and determine the arrest of the penetration through the cell membrane. With the advent of the Coronavirus Disease (COVID)-19 pandemic, it has been suggested that Sumac could be a possible antiviral agent against Severe Acute Respiratory Syndrome Coronavirus (SARS-CoV)-2. SARS-CoV-2 shares some similar structure and functional features with viruses as influenza, HIV, and HBV, towards which Sumac appears to act, as a lipidic envelope, affecting the infectivity action and other symptoms. Furthermore, some of the clinical manifestations of the SARS-CoV-2 infection concern the hemolytic and coagulative syndromes associated with the virus. The disruption of the erythrocyte membrane, the increase of ROS, and the decrease of antioxidant factors in these cells could contribute to vascular endothelial dysfunction and disseminated coagulation in the peripheral vases. The tannins of the Sumac could be incorporated into the erythrocytes membrane and promote resistance to pathogenic challenges that would be represented by the virus infection. Additionally, treatment in vitro or in vivo with Sumac compounds showed a stronger capacity to increase blood circulation, preventing the formation of coagulation factors [39]. Moreover, the anti-inflammatory and immunomodulatory activity of Sumac could be exploited to treat one of the best-known manifestations of SARS-CoV-2 infection, the so-called cytokine storm. It is linked to the hyperinflammatory acute systemic syndrome and dysregulation of the immune system orchestrated by virus infection, which directly impacts acute respiratory distress syndrome and multi-organ failure. Additionally, the immune system, which is suppressed in the first phases of the infection, becomes hyperactivated, contributing to cytokine production [41]. Docking studies about the phytochemicals of Sumac showed an affinity between both Hinokiflavone and Myrecetin, two major compounds of Sumac extracts, and the binding site of enzyme proteases involved in virus entry and replication, suggesting a possible inhibiting effect of the Sumac compounds in the replication of SARS-CoV-2 [42]. In vitro studies about this possible action of Sumac compounds are still lacking, but given the known Sumac properties, a possible role in inhibiting the spread of COVID-19 may be suggested.

The antifungal activity of Sumac extracts has not been widely discussed in the literature. Methanol extracts, particularly coriorianaphthyl ether, coriariaoic acid, and coriarianthracenyl ester, act against Aspergillus flavus and Candida albicans, while an alcohol extract and chloroform extract of *Rhus coriaria* possessed antifungal activity against Aspergillus niger (at the concentrations of 25, 50, 100, and 200 μg/mL) [43,44].

## 5. Treatment of Age-Related Diseases with Sumac Extracts

Sumac composition contributes to the modulation of several diseases, and it is possible to suggest a role in the contrast of age-related diseases such as cardiovascular diseases, diabetes, cancer, and microbial infections [11,19,39,45,46].

Age-related diseases are characterized by a profound decline in the clinical condition of the patients, which occurs with advanced age and often causes the assumption of a lot of drugs. At the basis and as a consequence of the aging decline, there is the increased senescence of cells, which induces a low-grade chronic status of inflammation, called “inflammaging”, which influences the onset and the worsening of the diseases [47,48].

The aim of modern aging research is to reduce the factor determining an increase in the risk of developing age-related disease manifestation and so ameliorate the health span of older people.

Type 2 diabetes is related to increased inflammation status and oxidative stress, with a particular presence of senescent cells in pancreatic islands [49]. Some data showed the capacity of Sumac extracts to reduce glucose levels and oxidative stress, ameliorating diabetic complications. Indeed, the use of methanol extracts on rats showed a delay in the onset of hyperglycemia, with a major sensitivity toward insulin action, after administration of 200 mg/Kg and 400 mg/Kg per day for 5 weeks [50]. The use of Sumac to reduce the serum levels of total cholesterol, triglycerides, low density lipoproteins, and blood glucose levels have been extensively demonstrated on animal models at several concentrations, i.e., from 50 µg/mL for the anti-diabetic activity of methanol extracts, to 150–500 µg/mL for anti-lipidemic properties of Sumac leaf extracts, until the direct consumption by broiler chicks, but relatively few studies have been done on humans [1,45,51]. Recently, clinical trials conducted on patients with a high or mild risk of cardiovascular disease or with diabetes, and on obese adolescents, showed that the consumption of 3g per day of Sumac powder for 3 months, or 500 mg/twice daily for 4 weeks, did not cause toxicity and had a positive effect on lipidemic markers, blood pressure, endothelial function, body mass index, and on the expression of both Apolipoprotein (Apo) B and ApoA-I in diabetes type II patients [19,45]. Thus, Sumac extracts could be considered a possible treatment for the metabolic syndrome, characterized by overweight, hypertriglyceridemia, hyperglycemia, and hypertension [52].

Given the content of polyphenols, known for the antioxidant property, a neuroprotective effect of Sumac powder has also been suggested. Neurodegenerative diseases often present cellular dysfunction, given by ROS action on DNA and proteins, or altered genetic factors involved in maintaining cellular homeostasis. Particularly, the capacity to polarize THP-1 cells to macrophage M1 phenotype, and to reduce levels of hydrogen peroxide in fibroblasts in Parkinson’s patients and in monocyte-derived macrophages THP-1, suggested a possible role of Sumac extracts at the concentrations of 3.125 μg/mL, 12.5 μg/mL, or 50 µg/mL to treat Parkinson’s diseases in terms of cellular dysfunction in the presence of a PARK2 mutation in a dose-dependent manner [53].

Experiments have also been made regarding the use of Sumac as an anti-cancer agent. Concerning this, the capacity of Sumac (100–150–200 µg/mL) to downregulate IL-6, IL-8, and TNF-α through NF-κB and Signal Transducer and Activator of Transcription 3 signalling in breast cancer cells has been shown. It has also demonstrated its activity in the modulation of nitric oxide pathways and the capacity to block migration (100–150–200 µg/mL) and invasion of triple negative breast cancer cells (10–50 µg/mL). Sumac extracts may also suppress angiogenesis and cell proliferations, affecting metalloproteinases 9 (50–100 µg/mL), prostaglandin E2 (150–200 µg/mL), and the adhesion to extracellular matrix components (100–200 µg/mL) in a concentration-dependent manner [46].

Furthermore, Sumac probably has anti-proliferative and pro-apoptotic effects on cancer cells acting on cell cycle checkpoints, such as p53 and p21, at the concentration range of 200 to 600 µg/mL [54]. On HT-29 and Caco-2 human colorectal cancer cells, its extracts induced apoptosis and autophagy by blocking mTOR/AKT pathways at increasing concentrations of 100–150–450 and 600 µg/mL, promoting the proteasome degradation of proteins after 24–48 h from the treatment [55].

These characteristics of Sumac make it a powerful candidate for the treatment of human diseases commonly found in the aged population, but also suggest that further studies must be conducted to define its properties and its possible role as a protective agent against unsuccessful aging. This concept is related to the employment of anti-aging molecules, i.e., natural compounds, drugs, or factors which have not only an application in the increase of the lifespan, but also an application in the rise of health span. They can act on serological markers, on molecular pathways, or may be involved in the regulation of metabolic processes [56]. Indeed, the ability of Sumac extracts to determine the cell cycle arrest and apoptosis in damaged cells may have an implication in diseases characterized by cellular dysfunction, such as senescent cells, favoring the definition of the role of Sumac as an anti-aging agent.

## 6. Application of Sumac Extracts in Aging Biomedicine

Notions about the nutraceutical properties of Sumac powder or its byproducts to treat age-related disturbances opened the door to the realization of novel therapeutic systems and biomedical tools.

The first consideration regards the potential role of Sumacs in the field of functional foods. The content in bioactive compounds, able to act on several aspects of health such as inflammation, oxidative stress, and infectious disease, is the main argument to evaluate its potential application in aging biomedicine. In fact, little is done about the direct utilization of Sumac products on humans, although several studies have demonstrated the safe use of this plant as a spice.

The use of Sumac for the treatment of several diseases and age-related pathologies, at the basis of which there is inflammatory or oxidative distress and an involvement of the senescent phenotypes, should be considered. These conditions could be treated at a single level or by evaluating the syndrome in its complexity, to restore the homeostatic condition, disturbed with advancing age.

As mentioned in the previous paragraphs about the anti-inflammatory and antioxidative characteristics of Sumac extracts, the treatment of inflammatory diseases demonstrated on in vitro models can be studied in depth for the realization of innovative drugs for humans [5,17,31,32,33]. Moreover, experiments on model rats demonstrated the antinociceptive action of Sumac extracts after treatment with acetic acid to reduce the pain as well as the complete and rapid wound healing [5]. The preparation of drugs based on natural compounds have to take into consideration not only the active principles present in the drugs, but also the additives and the carrying of the compounds used, which may interfere with the properties of the drug itself. Nanosystems, particularly nanoparticles, a form of drug delivery system, arise as prominent actors in the future of pharmacological synthesis. In this context, a study based on the realization of gold and silver nanoparticles demonstrated enhanced antimicrobial activity of silver nanoparticles fulfilled by Sumac extracts (25mL of water sumac extracts with 200mL of 1 mm of silver solution) toward pathogenic microorganisms. Silver nanoparticles, despite the contribution of plant extracts, caused denaturation of bacteria membranes as a result of the interaction between silver and enzymes of bacteria membrane, resulting in cell death. The inclusion of the Sumac extracts produced a stronger activity against pathogens, and furthermore, preserved DNA from the damage induced by oxidative stress. Comparing the effects of water extracts of Sumac with that of nanoparticles supplemented with plant extracts, the latter produced a stronger effect. Thus, silver and gold nanoparticles may have proved to be promising tools against fungal, viral, and inflammatory insults when coupled with natural compounds obtained by plant extracts [57]. In addition, this kind of drug delivery is useful in reducing the environmental and economic impact of drug production. Nanoparticles or the extracts could be used to realize pharmaceutical formulations such as sprays. This method could allow the preservation of the volatile components of Sumac extracts that are obtained from fruits or other Sumac parts. This formulation requires the use of drug carrying, for example maltodextrins, but may influence the Sumac properties that impact the bioactive compounds composition, manifesting a reduction of phenols and therefore the antioxidant property of the plant extract [58]. The main goal could be to develop alternative methods for preparing the extracts, including the use of green solvent, as solvent obtained by biomass and preparing systems with characteristics that facilitate their action.

Most of the studies reported were conducted using Sumac extracts, but the same or similar beneficial effects through the direct intake of the fruits or plant dried powder have been found. Considering the biodisponibility and bioactivity of 101–102 µg/mL of compounds in dried Sumac parts, it was already determined that Sumac could be used as a food fortifier. It was used in the preparation of yayik butter, a churned and soured yogurt from Turkey prepared with the addition of water and salt, in which 0.2–0.5% of *w*/*w* Sumac extracts resulted in an increase of shelf life of the product [5]. An Italian study showed how goat milk yogurt, supplemented with Sumac (20% *w*/*v*), has led not only to an increase of organoleptic and quality characteristics of the products, but also to an increase in the content of bioactive compounds, elevated phenols and tannins, and an increase in antioxidant activity, important for the food preservation [59]. In a recent study, Staghorn Sumac (*Rhus typhina*), the American Sumac, was used at different concentrations (less than 2% to 10%) as a supplement in bread making. The aim was to verify its properties in extending bread shelf life and increasing nutraceutical properties, determined by the higher polyphenol content of the Sumac used. The results showed that the use of Sumac at concentrations lower than 0.4% had less impact on the organoleptic properties of bread, reducing certain characteristics such as texture, color, and flavor, but increasing its polyphenol and anthocyanin content. This study opens new perspectives in the use of Sumac for the preparation of bread-based functional foods, which is widespread in Mediterranean diets, and is eaten by most of the elderly population [60]. Furthermore, in an Iranian clinical trial, 1.5g of Sumac 2 times/day together with pomegranate juice (200mL for 3 times/day) was administered to a COVID-19 cohort of patients and compared to a placebo treatment. Both Sumac and pomegranate, thanks to their high content in polyphenols, anthocyanins, and tannins, reduced the symptoms of COVID-19 patients (fever, chills, cough, smell and taste disorders, shortness of breath, diarrhea, nausea, vomiting, and abdominal pain), regardless of standard treatments [61]. Thus, the use of Sumac in foods or as a spice, has healthy benefits in reducing the symptoms of COVID-19 infection, probably acting as an antiviral and anti-inflammatory agent.

## 7. Conclusions

The rise in the average age worldwide will lead to a concomitant increase in the incidence of age-related diseases with inflammatory backgrounds. There has been an increase in research focused on the possibility of delaying the typical inflammatory burden of aging, called inflammaging, and the increased incidence of infectious diseases. The growing issue of antibiotic resistance and collateral effects due to the assumption of drugs, expanded the market for supplements. Nonetheless, the attention to sustainability is a key point worldwide.

In this context, Sumac is a promising plant with proven nutraceutical effects that are not yet fully tested in humans. It seems to have antimicrobial properties other than anti-inflammatory and antioxidant effects. Some Sumac compounds, such as quercetin and kaempferol, may also modulate the immune response, acting on the maturation and activation of dendritic cells, which can influence T cells, macrophages, and neutrophils activity.

The use of supplements obtained by phytochemicals extracted from Sumac could contribute to the treatment of inflammatory or infectious-based diseases in the elderly, such as upper and lower airway diseases and gastro-intestinal diseases. Rising knowledge about the immunoceutical properties of Sumac extracts could allow their use in the treatment of immune-mediated pathologies, typical of the aging process, considering the possibility of stimulating the immune cells to fight pathogenic challenges.

The phytochemicals of Sumac and their potential biomedical properties are the results of the climate, soil, and human intervention in defining the cultivation, harvesting, and processing techniques of the plant derivatives. Thus, choosing the extraction method of the bioactive compounds has a prominent role in the possible applications of Sumac in the biomedical field, because it has been proven that different kinds of extracts have different abilities. The aqueous and alcoholic extracts are more efficient in the modulation of the inflammatory, oxidative, and microbial effects thanks to the greater content in polyphenols. Therefore, the study of alternative methods to prepare the extracts, innovative drug delivery systems, and different formulations may not only facilitate the compliance of the patients and the efficacy of the drugs, but also introduce an incentive to green synthesis and innovation in biomedical and pharmacological research.

In conclusion, the importance that using Sumac could have in aging research therefore includes the possibility of easily obtaining the plant and its extracts, of being able to evaluate the use of new nanoparticle drug delivery systems, and the creation of alternative and more efficient pharmaceutical formulations. The latter could include spray formulations for respiratory tract diseases, or food and drink supplements, and could be created based on pathological characteristics to increase the effectiveness of the treatment.

## Figures and Tables

**Figure 1 ijms-24-06206-f001:**
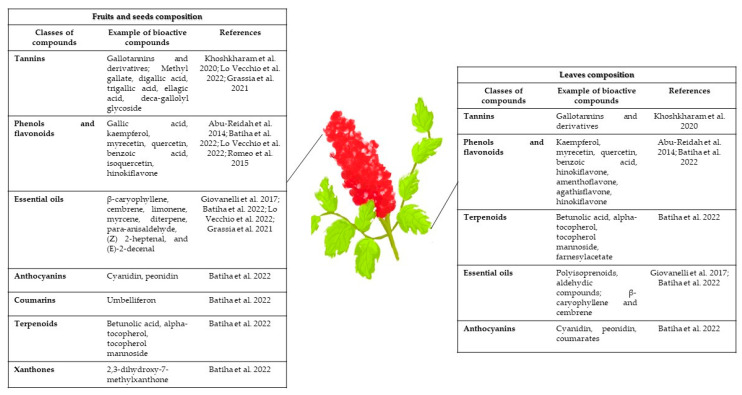
Phytochemicals present in Sumac shrub [1,2,6,8,11,14,15].

**Table 1 ijms-24-06206-t001:** Bioactive compounds of Sumac plants, according to extraction methods.

Compound	Method of Isolation	Concentration	Method of Detection	References
Fibers	Chemical methods codified by Association of Official Analytical Chemist	33.21 ± 1.02%	Chemical methods codified by Association of Official Analytical Chemist	[11]
Unsaturated fatty acids	Chemical methods codified by Association of Official Analytical Chemist	(65.09 ± 1.67%)	Chemical methods codified by Association of Official Analytical Chemist	[11]
Palmitic acid	Chemical methods codified by Association of Official Analytical Chemist	31.25 ± 0.47 mg/Kg	Chemical methods codified by Association of Official Analytical Chemist	[11]
Linolenic acid		1.85 ± 0.07%		
α-linoleic acid		30.82 ± 1.21%		
Polyphenols	Ethanol/water extraction	27.16±0.31 g gallic acid equivalents (GAE)/Kg (leaves); 5.34±0.43 g GAE/Kg (fruits)	High Performance Liquid Chromatography (HPLC)	[11]
Flavonoids				
Quercetin 3-O-galactoside	Methanol/aqueous extraction	160.53 ± 0.02 mg/g	HPLC	[11]
Kaempferol 3-O-glucoside		99.86 ± 0.01 mg/g		
Quercetin		23.13 ± 0.02 mg/g		
Myricetin 3-O-hexoside		18.55 ± 0.01 mg/g		
Organic acids				
Gallic acids	Methanol extracts	142.549 ± 0.02 mg/g	HPLC	[11]
Pentagalloyl-hexoside		128.09 ± 0.01 mg/g		
Malaric acid	Acqueous extraction	1568.04 ± 0.05 mg/Kg	Spectrometric and HPLC analysis	[8,14]
Citric acid		56.93 ± 0.35 mg/Kg		
Fumaric acid		3.40 ± 0.46 mg/Kg		
Tartaric acid		2.15 ± 0.13 mg/Kg		
Essential oils				
β-caryophyllene	Solid Phase Micro-Extraction	54.5%	Gas Chromatography/Mass Spectrometry (MS)	[7,12]
α-pyrene		15.2%		
Minerals				
Potassium	HNO_3_ and H_2_O_2_ warm extraction	266.91 ± 15.55 mg/Kg	Inductively coupled plasma MS	[11]
Calcium		215.53 ± 16.78 mg/Kg		
Magnesium		41.870 ± 3.55 mg/Kg		
Phosphorus		39.70 ± 3.05 mg/Kg		
Vitamins				
Pyridoxine	Water extraction	69.83 ± 0.31 mg/Kg	HPLC	[5,13]
Ascorbic acid		38.91 ± 0.27 mg/Kg		
Thiamine		30.65 ± 0.57 mg/Kg		
Riboflavin		24.68 ± 0.42 mg/Kg		

**Table 2 ijms-24-06206-t002:** Composition of Sumac components based on extraction solvent.

Type of Solvent Used for the Extraction	Part of Sumac Treated	Phytochemicals Composition	References
Water	Fruits	Organic acids: malic, citric, fumaric, and tartaric; Terpenoids; Tannins; Quinones; Sterol and steroids; Diterpenes; Phenols; Flavonoids; Anthocyanins; Proteins; Resines; Cardiac glycosides; Fatty acid: oleic acid, linoleic acid, and palmitic acid.	[5,8,15,16]
Water	Leaves	Phenols; Flavonoids: myricetin, quercetin; Tannins: gallotannins.	[5,17]
Ethanol	Fruits	Polyphenols; Flavonoids; Anthocyanins: cyanidin, peonidin, pelargonidin, petunidin, delphinidin glucosides and coumarates; Organic acids: malate, butanedioic acid; Terpenoids; Quinones; Sterol and steroids; Proteins and amino acids; Resines; Cardiac glycosides; Alkaloids; Oil and fatty acids; Tannins.	[5,6,15,16,18]
Ethanol and water	Fruits	Flavonoids; Anthocyanins; Tannins.	[17]
Methanol and water	Leaves	Tannins: gallotannins; Phenolic and Flavonoids derivatives: myricetin, quercetin.	[18]
Methanol	Fruits	Phenols; Hydrolysed tannins: gallotannins derivatives; Flavonoids; Anthocyanins; Butein.	[1,18]
Methanol	Leaves	Flavonols: quercetin, myricetin, and kaempferol; Tannins; Flavonols: quercetin, myricetin, and kaempferol; Organic acids: gallic acid, methyl gallate, m-digallic acid, and ellagic acid.	[5]
Ethyl acetate	Leaves	Flavonols: quercetin, myricetin, and kaempferol; Organic acids: gallic acid, methyl gallate, m-digallic acid, and ellagic acid.	[1]
Petroleum ether extract	Fruits	Organic acids: oleic acid, linoleic, palmitic, stearic acids and other fatty acids.	[1]

## Data Availability

Not applicable.

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
