# Peer review of "The Nutraceutical Properties of Rhus coriaria Linn: Potential Application on Human Health and Aging Biomedicine"

_ijms, 2023, doi:10.3390/ijms24076206_

Round 1

Reviewer 1 Report

In recent years, 8 review papers have been published with similar theme as below. Also, only 136 papers are found in Pubmed. So, this reviewer feels this paper has lack of novelty.

1: Batiha GE, Ogunyemi OM, Shaheen HM, Kutu FR, Olaiya CO, Sabatier JM, De Waard M. <i>Rhus coriaria</i> L. (Sumac), a Versatile and Resourceful Food Spice with Cornucopia of Polyphenols. Molecules. 2022 Aug 14;27(16):5179.

2: Hashem-Dabaghian F, Ghods R, Shojaii A, Abdi L, Campos-Toimil M, Yousefsani BS. Rhus coriaria L., a new candidate for controlling metabolic syndrome: a systematic review. J Pharm Pharmacol. 2022 Jan 5;74(1):1-12.

3: Mohit M, Nouri M, Samadi M, Nouri Y, Heidarzadeh-Esfahani N, Venkatakrishnan K, Jalili C. The effect of sumac (Rhus coriaria L.) supplementation on glycemic indices: A systematic review and meta-analysis of controlled clinical trials. Complement Ther Med. 2021 Sep;61:102766.

4: Ghafouri A, Estêvão MD, Alibakhshi P, Pizarro AB, Kashani AF, Persad E, Heydari H, Hasani M, Heshmati J, Morvaridzadeh M. Sumac fruit supplementation improve glycemic parameters in patients with metabolic syndrome and related disorders: A systematic review and meta-analysis. Phytomedicine. 2021 Sep;90:153661.

5: Alsamri H, Athamneh K, Pintus G, Eid AH, Iratni R. Pharmacological and Antioxidant Activities of <i>Rhus coriaria</i> L. (Sumac). Antioxidants (Basel). 2021 Jan 8;10(1):73. 

6: Elagbar ZA, Shakya AK, Barhoumi LM, Al-Jaber HI. Phytochemical Diversity and Pharmacological Properties of Rhus coriaria. Chem Biodivers. 2020 Apr;17(4):e1900561.

7: Sakhr K, El Khatib S. Physiochemical properties and medicinal, nutritional and industrial applications of Lebanese Sumac (Syrian Sumac - <i>Rhus coriaria</i>): A review. Heliyon. 2020 Jan 27;6(1):e03207.

8: Akbari-Fakhrabadi M, Heshmati J, Sepidarkish M, Shidfar F. Effect of sumac (Rhus Coriaria) on blood lipids: A systematic review and meta-analysis. Complement Ther Med. 2018 Oct;40:8-12.

Reviewer 2 Report

The nutraceutical properties of Rhus coriaria Linn and potential applications in biomedicine were reviewed. Rhus coriaria Linn is a plant growing in the Mediterranean basin, including Sicily, where it is known as Sicilian Sumac. Since antiquity, it has been used as a medicinal herb, considering its pharmacological properties and its recognized anti-inflammatory, antioxidant, and antimicrobial effects. Multiple studies highlighted that the beneficial properties of Sumac extracts depend on the abundance of present phytochemicals such as polyphenols, fatty acids, minerals, and fibres. Despite its wide use as a spice, the literature is still scarce with documented studies on Sumac health effects on humans. The review is interesting, but it needs major revision according to the specific remarks written further.

Specific remarks

The title should be changes since it is not clear that Rhus coriaria and Sicilian Sumac are the same plant. In addition, there is need to state what range of years this review is covering. What about phytochemical composition that was reviewed and not mentioned in the title? Nutraceutical properties should be removed from the title.

The abstract should be changed. The sentence starting with “Moreover, the progressive increase of the mean age of the world population took to a major requirement…” is too general and not related specific to the present review. There is need to state the novelty of proposed review in the abstract.

At the end of introduction part there is need to clearly state if this is the first review on Rhus coriaria. If another reviews are available there is need to state the novelty in comparison with them. The range of years covered with present review should be mentioned.

In the paragraph 2. Sumac composition, besides listing the identified phytochemicals from this plant there is need to add the range of their determined quantities as well as the methods used for their isolation and identification (in the text).

Ian all reported parts of biological activities of Sumac there is need to add the concentration levels of the extract for noted biological activities.

The conclusion is too general. It should be changed completely. There is need to address the phytochemical composition and observed biological activities of Sumac and to give ideas for further research. Some general observations can be present, but the conclusions should be more focused on the review of Sumac.

Round 2

Reviewer 1 Report

Some are understanding, so it is now acceptable.

Reviewer 2 Report

The manuscript was revised according to the comments.